# Pulmonary Hemorrhage in Premature Infants: Pathophysiology, Risk Factors and Clinical Management

**DOI:** 10.3390/biomedicines13071744

**Published:** 2025-07-16

**Authors:** Sariya Sahussarungsi, Anie Lapointe, Andréanne Villeneuve, Audrey Hebert, Nina Nouraeyan, Satyan Lakshminrusimha, Yogen Singh, Christine Sabapathy, Tiscar Cavallé-Garrido, Guilherme Sant’Anna, Gabriel Altit

**Affiliations:** 1Division of Neonatology, Department of Pediatrics, Montreal Children’s Hospital, McGill University, Montreal, QC H4A 3J1, Canada; sariya.sahussarungsi@mail.mcgill.ca (S.S.); guilherme.santanna@mcgill.ca (G.S.); 2Division of Neonatology, Department of Pediatrics, CHU Sainte-Justine, Université de Montréal, Montreal, QC H3T 1C5, Canada; anie.lapointe@umontreal.ca (A.L.); andreanne.villeneuve@umontreal.ca (A.V.); 3Division of Neonatology, Department of Pediatrics, Centre Hospitalier de l’Université Laval, Université Laval, Québec, QC G1V 0E8, Canada; audrey.hebert.2@ulaval.ca; 4Jewish General Hospital, McGill University, Montreal, QC H4A 3J1, Canada; nina.nouraeyan@mcgill.ca; 5Division of Neonatology, Department of Pediatrics, University of California, Davis, Sacramento, CA 95817, USA; slakshmi@ucdavis.edu (S.L.); ygsingh@ucdavis.edu (Y.S.); 6Division of Hematology, Department of Pediatrics, Montreal Children’s Hospital, McGill University, Montreal, QC H4A 3J1, Canada; christine.sabapathy@mcgill.ca; 7Division of Cardiology, Department of Pediatrics, Montreal Children’s Hospital, McGill University, Montreal, QC H4A 3J1, Canada; tiscar.cavalle.med@ssss.gouv.qc.ca

**Keywords:** pulmonary hemorrhage, preterm infants, very low birth weight (VLBW), fetal growth restriction (FGR), respiratory distress syndrome (RDS), patent ductus arteriosus (PDA), surfactant therapy, coagulopathy, lung ultrasound, echocardiography

## Abstract

Pulmonary hemorrhage (PH) is a life-threatening complication predominantly affecting preterm infants, particularly those with very low birth weight (VLBW) and fetal growth restriction (FGR). Typically occurring within the first 72 h of life, PH is characterized by acute respiratory deterioration and significant morbidity and mortality. This review synthesizes current evidence on the multifactorial pathogenesis of PH, highlighting the roles of immature pulmonary vasculature, surfactant-induced hemodynamic shifts, and left ventricular diastolic dysfunction. Key risk factors include respiratory distress syndrome (RDS), hemodynamically significant patent ductus arteriosus (hsPDA), sepsis, coagulopathies, and genetic predispositions. Diagnostic approaches incorporate clinical signs, chest imaging, lung ultrasound, and echocardiography. Management strategies are multifaceted and include ventilatory support—particularly high-frequency oscillatory ventilation (HFOV)—surfactant re-administration, blood product transfusion, and targeted hemostatic agents. Emerging therapies such as recombinant activated factor VII and antifibrinolytics show promise but require further investigation. Preventive measures like antenatal corticosteroids and early indomethacin prophylaxis may reduce incidence, particularly in high-risk populations. Despite advancements in neonatal care, PH remains a major contributor to neonatal mortality and long-term neurodevelopmental impairment. Future research should focus on individualized risk stratification, early diagnostic tools, and optimized treatment protocols to improve outcomes. Multidisciplinary collaboration and innovation are essential to advancing care for this vulnerable population.

## 1. Introduction

Pulmonary hemorrhage (PH) is generally described as an acute event characterized by the sudden discharge of bloody fluid from the upper respiratory tract or through the endotracheal tube [1]. PH is a severe and often life-threatening condition primarily affecting premature infants, particularly those with very low birth weight (VLBW) [2] and intrauterine growth restriction (FGR) or those who are small for gestational age (SGA) [3]. It mainly occurs during the first 48 to 96 h of life, i.e., during the transitional period [4]. PH is characterized by rapid clinical deterioration, often leading to acute hypoxemia and respiratory failure. Its occurrence is strongly associated with several neonatal complications, including birth asphyxia and hypothermia [5], respiratory distress syndrome (RDS) [6,7,8], sepsis [9,10], and patent ductus arteriosus (PDA) [8,10] (Figure 1). The introduction of surfactant therapy has significantly improved neonatal outcomes in RDS management, yet paradoxically, it has been linked to an increased risk of PH in certain high-risk populations [11,12]. Despite advances in neonatal care, PH remains a major cause of mortality and long-term morbidity in preterm infants [9,12]. Over the past few decades, our understanding of PH in preterm infants has evolved from a narrow focus on vascular fragility to a broader recognition of its multifactorial pathophysiology. Contributing factors include pulmonary hemodynamic disturbances, surfactant-induced changes, cardiac immaturity, prenatal influences, impaired left-sided cardiac compliance, and genetic or inflammatory predispositions. Despite these insights, significant gaps remain—particularly in standardizing diagnostic criteria, clarifying the interplay among known risk factors, and integrating early detection tools such as lung ultrasound, heart rate variability analysis, and targeted neonatal echocardiography. Additionally, effective prevention and management strategies are yet to be clearly defined. This narrative review synthesizes current evidence, addresses key diagnostic and therapeutic challenges, and proposes future directions to improve outcomes in this vulnerable population.

## 2. Epidemiology

PH is a major complication in very premature infants, particularly those with very low birth weight (VLBW, <1500 g) and extremely low birth weight (ELBW, <1000 g). Its overall incidence ranges from 1 to 12 per 1000 live births [6] but increases to approximately 11.9% [13] in VLBW infants. The associated mortality is up to 57% [14]. While regional variations are likely, detailed geographic data are limited. In high-income settings, PH is more frequently reported, reflecting broader access to neonatal intensive care and interventions such as surfactant therapy and high-frequency ventilation. In low-resource settings, limited access to NICUs and specialized care leads to the underreporting of PH and higher mortality rates due to delayed interventions [15]. A 2012 systematic review reported a PH incidence in high-income countries ranging from 1 to 12 per 1000 live births [4]. A study from China found PH in 6.6% of infants born with <1500 g and 22.9% in infants with BW <1000 g [16]. Variability in incidence and outcomes is influenced by differences in the definition of PH and care practices, including the timing and frequency of surfactant administration, antenatal steroid use, and the availability of advanced ventilatory support. Regional studies underscore PH’s contribution to neonatal mortality: in Taiwan, PH reportedly accounted for 10.5% of deaths among VLBW infants, with one multicenter study reporting a rate between 2.2 and 27.6% across hospitals [17]. Similarly, a Korean tertiary center reported rates of 20.5% in ELBW and 14.8% in VLBW infants [13].

Demographic factors also influence PH incidence. Male infants have a higher incidence than females [18]. Access to prenatal care reduces the risk of prematurity and PH [19], while socioeconomic disparities increase vulnerability through poorer maternal health and limited access to prenatal and neonatal care [15]. Data on ethnic differences in PH remain limited, emphasizing the need for further research.

## 3. Pathogenesis

The pathogenesis of PH in neonates is multifactorial, driven by both the structural and functional immaturity of the pulmonary vasculature. Similarly to intraventricular hemorrhage, PH primarily affects premature infants whose fragile capillaries are highly susceptible to rupture under fluctuating hemodynamic conditions. (Figure 2) During the immediate postnatal transition, the physiological drop in pulmonary vascular resistance (PVR) leads to a sudden increase in pulmonary blood flow. In preterm infants, this surge in blood flow can overwhelm underdeveloped vascular structures. In this context, a combination of vascular fragility, immature autoregulatory mechanisms, immature cardiac function, genetic predisposition, and iatrogenic factors—such as positive pressure ventilation and fluid shifts—can exacerbate capillary stress, precipitating hemorrhagic episodes. Emerging evidence suggests that hemodynamic conditions and genetic predispositions [20] may further modulate individual susceptibility to PH, emphasizing the need for personalized approaches to risk stratification and management.

The degree of prematurity is the most significant risk factor for PH, with infants born before 30 weeks of gestation are particularly vulnerable. The structural immaturity of the pulmonary capillary network in these infants results in increased fragility, making it more prone to rupture under stress [17,21]. This increased risk is attributed to a combination of immature lung development, the frequent presence of RDS requiring surfactant administration, and the inherent fragility of the pulmonary capillary network. As birth weight decreases, the risk of PH rises significantly [22].

Platelet dysfunction is recognized as another contributing factor to pulmonary hemorrhage in preterm neonates, particularly among those with very low birth weight (VLBW). In this population, bleeding events most commonly occur within the first week of life, which corresponds to the period when platelet dysfunction tends to be the most pronounced [23]

## 4. Factors Contributing to Pulmonary Hemorrhage

### 4.1. Respiratory Distress Syndrome (RDS)

Extremely premature infants are born at a critical stage of pulmonary development, where alveolarization is incomplete, surfactant production has not started or remains insufficient, and airways, as well as pulmonary vascular structures, are still immature.

During the course of RDS, infants with surfactant deficiency are at high risk for alveolar edema due to the accumulation of fluid and erythrocytes within the alveoli. This accumulation increases alveolar epithelial permeability, promoting infiltration by immune cells and proinflammatory cytokines, particularly neutrophils, which are key mediators of the innate immune response. Over time, the disruption of intercellular junctions leads to epithelial barrier breakdown, allowing for the further migration of fluid and red blood cells into air spaces, further compromising gas exchange [24,25].

The rapid improvement in lung compliance with surfactant exposure can lead to a sudden reduction in PVR, abruptly increasing pulmonary blood flow. This sudden hemodynamic and compliance change increases the risk of capillary rupture and the loss of epithelial and endothelial integrity and bleeding, precipitating PH (Figure 3) [17,26,27]. A meta-analysis of surfactant therapy published in 1993 reported a modest increase in the risk of pulmonary hemorrhage [28]. However, this study predates the routine survival of extremely preterm infants born at <25 weeks of gestation and may not be generalizable to the current population. Updated studies in this specific subgroup are needed to clarify the risk. However, this increased risk is outweighed by the substantial benefits of surfactant administration in reducing mortality in preterm infants with severe RDS. In a comparative study, the incidence of PH was significantly higher in infants treated with poractant-alpha (200 mg/kg) than those receiving beractant (100 mg/kg) (14.3% vs. 0.0%, *p* = 0.038) [29]. Similarly, Ahmad et al. reported that newborns with PH had greater cumulative exposure to surfactant therapy, particularly poractant alfa, compared to matched controls [14]. Earlier studies on poractant alfa did not demonstrate this association [30,31], likely due to the limited inclusion of extremely preterm infants (23 or 24 weeks of gestation)—those at the highest risk for PH. The surfactant may also interfere with the coagulation cascade. An in vitro study by Strauss et al. showed that higher surfactant concentrations were associated with a trend toward prolonged clotting times and reduced clot strength. While the findings suggest a potential exacerbation of bleeding, the study was conducted in adult populations, and it remains uncertain whether it can be extrapolated to the neonatal population [32].

Finally, mechanical ventilation can also exacerbate the risk of PH. High airway pressures and excessive tidal volumes can cause alveolar overdistension, leading to capillary rupture through volutrauma and barotrauma. Additionally, oxygen-induced lung injury can amplify pulmonary vascular stress, leading to endothelial damage and increased capillary permeability [33].

### 4.2. Patent Ductus Arteriosus (PDA) and Pulmonary over Circulation

The contribution of a patent ductus arteriosus to the pathogenesis of pulmonary hemorrhage remains controversial. In immediate postnatal life, the physiological drop in PVR driven by birth, surfactant therapy with fluctuations in airway pressure, oxygen exposure, hypocapnia, and the use of vasodilatory agents such as iNO and opioids may facilitate left-to-right shunting through unrestrictive PDA, overwhelming the fragile pulmonary vascular bed. This shunt exposes the immature pulmonary vasculature to systemic pressure and elevated blood flow (an increased pulmonary-to-systemic flow ratio [Qp/Qs]) potentially exacerbating pulmonary edema and the risk of capillary rupture [4,24]. It is not clear if early extubation after surfactant therapy (INSURE) or less invasive surfactant administration (LISA) is associated with a higher incidence of PH compared to continued mechanical ventilation after surfactant administration. Centers that use high-frequency ventilation and early hemodynamic treatment for PDA report a lower incidence of PH (Figure 3). However, as outlined in a recent meta-analysis [34] and a review by our group [35], the early use of medications to accelerate ductal closure in extremely preterm infants does not appear to confer short- or long-term benefits. This practice has been increasingly discouraged, including in the most recent guidelines from the American Academy of Pediatrics [36].

Despite these theoretical concerns, clinical trials have yielded conflicting evidence regarding the role of PDA in PH. In a randomized controlled trial (RCT) of infants <29 weeks, early indomethacin administration significantly reduced PH in the first 72 h (1% vs. 21%) but not over the entire study period (9% vs. 23%, *p* = 0.07) [37]. The observed early benefit may reflect the direct pulmonary vasoconstrictive effect of NSAIDs [38] or anti-angiogenic properties [39] rather than PDA constriction itself.

Further evidence challenges the centrality of PDA in PH pathogenesis. The Trial of Indomethacin Prophylaxis in Preterms (TIPPs) found no difference in PH incidence between infants receiving prophylactic indomethacin (15%) and a placebo (16%) [40]. In a recent cohort study using a conservative PDA approach (no NSAIDs, acetaminophen, surgical ligation, or catheter-based closure), PH rates were low: 6% in preterm infants born <26 weeks of GA (n = 130) and 7% in those born between 26 and 28 + 6 weeks (n = 84) [41]. Similarly, the BeNeDuctus trial reported a PH rate of 3% in the expectant management group versus 1% in the ibuprofen group [42]. Notably, this trial included a true non-intervention control group, with only one participant in the expectant group receiving open-label treatment (ibuprofen). These findings suggest that PDA might contribute to the pathogenesis of PH but challenge the hypothesis that it is a primary driver of that. Instead, PH appears to be more strongly influenced by intrinsic factors related to the immature pulmonary vascular bed and pulmonary venous drainage along with the disruption of the alveolar epithelial barrier rather than the persistence of ductal shunting.

### 4.3. Stress Failure of Pulmonary Capillaries

The stress failure of pulmonary capillaries leads to the disruption of the endothelial barrier, allowing hemorrhagic fluid to leak into the alveoli. This process is driven by three primary forces: (1) circumferential tension within the capillary wall caused by transmural pressure across the capillary; (2) surface tension within the alveoli, which stabilizes the bulging capillaries; and (3) longitudinal tension in the alveoli, which is generated during lung inflation [4,43]. Sudden reductions in PVR, venous congestion, or left heart dysfunction can abruptly alter the pulmonary vascular flow, subjecting the fragile capillary beds of preterm infants to excessive mechanical stress. This increased stress can ultimately result in capillary rupture and hemorrhage (Figure 3A–C).

### 4.4. Left Ventricular Stiffness and Diastolic Properties

Emerging evidence suggests that cardiac function and the stress–velocity relationship, a surrogate marker of myocardial workload and afterload mismatch, may play a role in the pathogenesis of pulmonary hemorrhage (PH) in preterm infants [44]. The stress–velocity relationship reflects the dynamic interaction between myocardial contractility and wall stress, integrating both systemic vascular resistance and ventricular performance. In the transitional period, preterm infants with immature myocardium and labile hemodynamics are particularly vulnerable to fluctuations in afterload. A sudden increase in systemic vascular resistance, such as that caused by vasopressor use, can disrupt the normal stress–velocity relationship of the immature myocardium. This afterload-intolerant state impairs left ventricular performance, leading to elevated left atrial and pulmonary venous pressure. The resulting capillary stress predisposes fragile pulmonary vessels to leakage and hemorrhage. This mechanism is analogous to the cerebral vulnerability seen in intraventricular hemorrhage (IVH) [44,45]. The stress–velocity relationship may serve as a hemodynamic biomarker linking ventricular performance to pulmonary capillary integrity, highlighting the need for the careful modulation of cardiovascular support in extremely low-birth-weight infants.

High-risk populations, such as infants with intrauterine growth restriction due to placental insufficiency [45], often exhibit systemic vascular stiffness and compromised LV compliance [46,47]. Similarly, neonates with twin-to-twin transfusion syndrome (TTTS) may present with LV hypertrophy, diastolic dysfunction, and elevated LA pressure. In these instances, the transpulmonary pressure gradient is reduced, promoting post-capillary congestion and increasing the possibility of capillary rupture and hemorrhage.

### 4.5. Genetic Factors

Hereditary hemorrhagic telangiectasia (HHT), or Osler–Weber–Rendu syndrome, is a rare but autosomal dominant disorder characterized by multisystem telangiectasias and arteriovenous malformations (AVMs). While adult complications are well-documented, neonatal presentations are rare and limited to case reports. In children, HHT typically presents with cyanosis, digital clubbing, and dyspnea, with or without visible telangiectasias. PH is uncommon in early life and typically emerges later due to progressive AVMs or bronchial angiodysplasia. The pathogenesis of PH in HHT is linked to impaired vascular integrity, driven by dysregulated transforming growth factor-β signaling, which disrupts endothelial cell function and extracellular matrix composition [48]. Although a rare condition, HHT and other genetic disorders should be considered, especially in the context of familial history, due to their potential to predispose infants to vascular integrity [20].

Another rare hereditary bleeding disorder is hemophilia, which follows an X-linked recessive inheritance pattern and may also appear atypically with PH as an initial symptom. Pace et al. reported a case involving a premature neonate from a monochorionic-diamniotic twin gestation who developed severe, life-threatening PH as the first clinical indication of hemophilia B despite an absence of any family history of bleeding disorders [49].

### 4.6. Infection and Sepsis

In neonates with overwhelming sepsis—particularly Gram-negative sepsis—PH may result from profound microvascular injury and inflammation. Endotoxin release activates mononuclear phagocytes, stimulating the production of proinflammatory cytokines, which, in turn, trigger neutrophils and T-cells to release secondary inflammatory mediators. This inflammatory surge disrupts endothelial integrity and increases pulmonary capillary permeability, heightening the risk of PH [17]. Additionally, sepsis induces a systemic inflammatory response that disrupts normal coagulation pathways, increasing the risk of disseminated intravascular coagulation (DIC). Case reports have also suggested associations between PH and conditions such as coxsackievirus infection, necrotizing enterocolitis, and early-onset sepsis, although data remain limited [14,50,51]. In preterm infants with immature pulmonary vasculature, the synergy between sepsis-induced microvascular permeability, endothelial damage, coagulopathy, and fragile capillaries creates a critical environment that is highly conducive to PH [17]. Sepsis is a common cause of pulmonary hemorrhage in low-resource settings.

Differentiating PH from severe sepsis remains clinically challenging, particularly when both conditions coexist as part of a systemic inflammatory response. Further research is needed to develop integrated diagnostic frameworks combining imaging, laboratory, and hemodynamic data to improve accuracy and guide management when there is diagnostic uncertainty between these two conditions.

### 4.7. Fetal Growth Restriction (FGR)

FGR is associated with an increased risk of PH, likely due to a combination of coagulopathy, thrombocytopenia, platelet dysfunctions, and cardiovascular dysfunctions. Impaired coagulation in these infants may involve reduced levels of vitamin K-dependent clotting factors, which is essential for coagulation. Thrombocytopenia is frequently observed, particularly in cases associated with congenital infections such as cytomegalovirus. While the exact etiology of this coagulopathy remains unclear, antenatal liver hypoxia has been suggested as a contributing factor, potentially due to the impaired production of clotting factors [52]. Cardiac dysfunction may also compound this risk. Autopsy studies by Takahashi et al. in FGR infants with PH and heart failure revealed myocardial fiber hypoplasia and depleted glycogen stores, which are findings indicative of chronic hypoxia and malnutrition in utero rather than acute ischemia. As a result, FGR infants may experience compromised cardiac function after birth, struggling with both left and right ventricular dysfunction [53]. In turn, impaired cardiac function and elevated left atrial pressure can exacerbate pulmonary venous congestion, contributing to the pathogenesis of PH (Figure 4).

### 4.8. Other Conditions Possibly Associated with PH

Congenital heart disease (CHD) associated with pulmonary hypertension presents a paradoxical state marked by both thrombotic and bleeding tendencies [54]. CHD is implicated in 30–40% of PH cases, underscoring its clinical relevance [55,56].

In asphyxiated neonates, left ventricular failure and the subsequent increase in pulmonary capillary pressure are key hemodynamic contributors [19] to PH. Asphyxia leads to bradycardia, acidosis, and impaired myocardial contractility due to hypoperfusion and ischemia [57]. This cascade elevates pulmonary venous pressures, increasing the risk of capillary stress failure and PH. Moreover, neonates with perinatal asphyxia frequently exhibit evidence of coagulopathy, which may increase the risk of PH [58].

Neonatal hypothermia has also been associated with PH, likely through platelet dysfunction, resulting in thrombocytopenia or decreased response, which may persist or even accelerate during rewarming. If a similar process occurs in vivo, it may compromise hemostasis and contribute to hemorrhagic complications [59].

Coagulation factors also play an important role in PH. Interestingly, coagulation levels are physiologically lower in neonates, with even more pronounced deficiencies observed in preterm infants. A study by Pal et al. demonstrated that these factors gradually mature and typically reach adult levels by approximately six months of age [60].

In addition, coagulation disorders are frequently observed in neonates with PH and may exacerbate the condition [4,19,61]. Carolis et al. suggested that hypocoagulability is a key risk factor strongly associated with PH. This is supported by findings that neonates with PH more frequently exhibit thrombocytopenia and abnormal coagulation test results compared to those without PH. These infants also show increased rates of bleeding from other sites, including umbilical bleeding, widespread oozing, and bruising [3]. In a study by Yum et al., a prolonged prothrombin time/international normalized ratio (PT/INR) was associated with worse survival, and platelet count influenced the interval between PH onset and mortality [19]. Although activated partial thromboplastin time (aPTT) values remained within the normal range, they were predictive of adverse outcomes in the context of DIC.

These findings highlight the multifactorial nature of PH (Table 1) in neonates, where cardiovascular compromise, coagulopathy, and systemic stressors converge to produce overwhelming pulmonary integrity.

## 5. Protective Factors

### 5.1. Antenatal Glucocorticoids

Berger et al. reported that the maternal administration of a complete course of antenatal glucocorticoids was associated with a reduced risk of PH compared to no glucocorticoids or an incomplete course [19]. This protective effect is likely due to the role of antenatal steroids in promoting lung maturation, enhancing surfactant production, and stabilizing the pulmonary vasculature, thereby reducing the risk of capillary stress failure and hemorrhage in preterm infants.

### 5.2. Prophylactic Indomethacin

The Trial of Indomethacin Prophylaxis in Preterm Infants (TIPP) demonstrated that prophylactic indomethacin did not reduce the overall incidence of PH. However, when analyzing its effect on severe pulmonary hemorrhage, the study found that it was associated with a lower incidence of early severe PH (within the first week of life) [62]. Despite this early benefit, its effectiveness in preventing late-onset severe PH (beyond the first week of life) was significantly reduced, suggesting that other mechanisms, beyond PDA-related hemodynamics, contribute to the pathogenesis of pulmonary hemorrhage in preterm infants [62].

## 6. Clinical Diagnosis

The diagnosis of PH in neonates is often urgent, requiring rapid recognition and intervention due to its association with sudden clinical deterioration. It is primarily based on clinical presentation, including acute respiratory distress, bloody tracheal secretions, and hypoxemia, supported by radiological findings such as diffuse alveolar infiltrates on chest radiography. Laboratory evaluations, including coagulation profiles and markers of inflammation, can help identify underlying contributing factors. In recent years, lung ultrasound (LUS) has emerged as a valuable, non-invasive diagnostic tool for PH. LUS can detect increased pulmonary echogenicity, consolidations, and alveolar–interstitial syndrome, aiding in the early identification of PH in the clinical context [63,64,65].

### 6.1. Clinical Presentation

PH typically occurs in the first 48 to 72 h of life. The hallmark clinical sign is the sudden appearance of frothy, pink-tinged secretions or visible bleeding from the endotracheal tube. In infants receiving non-invasive ventilation, blood may be observed in the posterior pharynx, or it may be detected in gastric secretions. This presentation is often accompanied by a rapid increase in oxygen requirements and ventilatory support, signaling a significant pulmonary compromise that requires urgent intervention. Lung auscultation often reveals coarse breath sounds and reduced air entry, reflecting pulmonary congestion and alveolar flooding. As PH progresses, oxygen requirements escalate [24] and may lead to apnea, widespread pallor, cyanosis, bradycardia, and hypotension due to hypovolemic shock if left untreated. In severe cases, rapid cardiopulmonary collapse can occur [1]. Given the acute and life-threatening nature of PH, clinicians must promptly recognize its onset, particularly in neonates with predisposing factors such as surfactant therapy, PDA, FGR, and coagulopathies. A sudden and unexplained clinical deterioration in a high-risk neonate should immediately raise suspicion for PH and prompt urgent evaluation and intervention [4,13,24,27].

### 6.2. Laboratory Diagnosis

Laboratory findings often reflect both the severity of blood loss and any underlying coagulopathy. A complete blood count (CBC) typically reveals a sharp drop in hemoglobin and hematocrit levels due to ongoing pulmonary bleeding, which may necessitate urgent blood transfusions to restore the oxygen-carrying capacity. Coagulation studies often demonstrate abnormalities, particularly in neonates with DIC or other coagulation disorders. Prolonged PT and aPTT, along with decreased fibrinogen levels and thrombocytopenia or platelet dysfunction, may exacerbate PH [4]. In addition, blood gas analysis frequently shows severe hypoxemia, hypercapnia, and metabolic acidosis, which are indicative of impaired gas exchange due to alveolar flooding and ventilation–perfusion mismatch.

### 6.3. Radiological Diagnosis

Chest radiography findings are often nonspecific and vary depending on the severity and timing of the hemorrhage. In mild cases, imaging may reveal fluffy opacities (Figure 5) or focal ground-glass opacities, reflecting localized alveolar hemorrhage. As the hemorrhage progresses, the opacities may become more diffuse, resembling worsening pulmonary edema. In severe cases, extensive alveolar flooding can lead to a complete “white-out” appearance (Figure 6), indicating a significant impairment of gas exchange [4]. However, these radiographic findings are not pathognomonic and can overlap with conditions such as RDS, pneumonia, and pulmonary edema, making clinical correlation essential for accurate diagnosis.

### 6.4. Lung Ultrasound in the Evaluation of PH

LUS is a promising bedside modality with potential utility in the diagnosis [65,66] and monitoring of PH in neonates. Its radiation-free, low-cost nature makes it particularly appealing in resource-limited settings. Preliminary studies suggest that certain LUS features—such as lung consolidation, air or fluid bronchograms, pleural effusion, B-lines, and shred signs—are frequently observed in neonates with established PH [65,66]. Liu et al. (2023) and Ren et al. (2017) describe in retrospective studies (n = 42 and n = 57, respectively) relatively consistent ultrasound findings in confirmed cases, with high reported sensitivities for signs like the shred sign (which refers to the irregular, jagged, or serrated border appearance of the pleural–lung interface described as the appearance of “torn paper”.) [65,66]. However, these findings are not specific to PH and may also occur in pneumonia or meconium aspiration. Importantly, in both studies, imaging was performed after the clinical diagnosis of hemorrhage, limiting conclusions about the predictive value or applicability for early detection and therapeutic guidance. These studies were also limited by small sample sizes, subjective diagnostic criteria, and the lack of generalizability to other clinical contexts—especially low-income settings. No trial to date has assessed whether LUS can meaningfully predict PH, support accurate diagnosis, or guide management in ways that improve outcomes such as survival, severe IVH, or pulmonary morbidity. Moreover, given the overlap of LUS findings with more benign conditions, premature adoption without robust evidence can lead to overdiagnosis, unnecessary treatment, or diagnostic anchoring, potentially causing harm. While LUS remains an appealing tool with significant potential, especially in low-resource settings, its integration into clinical pathways must be supported by prospective research and trials assessing its value in prevention, diagnosis, and therapeutic titration.

Characteristic LUS findings associated with PH include lung consolidation with a shred sign at the edges of the affected area, air or fluid bronchograms, and atelectasis, reflecting significant alveolar involvement. Additionally, abnormalities such as disrupted pleural lines, the absence of A-lines, and pleural effusion may be observed, with severe cases showing real-time evidence of red blood cell destruction and fibrous protein deposition, which appear as floating objects within the pleural fluid. Furthermore, alveolar–interstitial syndrome, characterized by the presence of more than three B-lines per intercostal space, suggests lung edema, which can be a consequence of hemorrhagic lung injury.

The concept of AI-assisted LUS interpretation holds promise for improving the early detection and standardization of PH diagnosis, especially in settings lacking expert imaging interpretation. If properly developed, these technologies could enhance diagnostic consistency, support earlier intervention, and reduce reliance on advanced imaging infrastructure. AI tools are still in their early developmental stages, and their application to PH requires collaborative work in bioengineering and neonatology. Prospective studies are essential to design, validate, and evaluate the accuracy and clinical impact of these technologies within diagnostic frameworks.

### 6.5. Echocardiography Assessment

#### 6.5.1. Patent Ductus Arteriosus

A study by Kluckow et al. examined echocardiographic findings in preterm infants around the time of PH onset and identified significant patterns associated with the condition [67]. Infants who developed PH had significantly larger PDA diameters (>1.5 mm), with a predominantly left-to-right shunt. Additionally, these infants exhibited an absent or retrograde diastolic flow in the post-ductal descending aorta, suggesting the presence of a steal effect from the systemic circulation and increased Qp/Qs.

#### 6.5.2. Pulmonary Hypertension

Inhaled nitric oxide (iNO) is approved for full-term and near-term infants with hypoxic–ischemic failure but has not demonstrated consistent outcome benefits in preterm infants [68,69]. However, some studies have seen potential oxygenation improvement in select preterm subgroups—especially those with pulmonary hypoplasia, a prolonged rupture of membranes, or early PPHN [70]. Observational studies have also indicated that an acute response to iNO in preterm infants is associated with improved survival, supporting the rationale for a short therapeutic trial in this population [71]. PPHN or acute pulmonary hypertension frequently complicates severe RDS in very preterm infants, and iNO administration is associated with oxygenation improvements and survival [72].

iNO may rapidly reduce pulmonary vascular resistance, leading to an abrupt increase in pulmonary blood flow, placing excessive stress on the immature and fragile capillary network, and increasing the risk of capillary rupture and PH. Moreover, abrupt iNO withdrawal may trigger rebound pulmonary hypertension, further stressing the pulmonary vasculature [73]. PH can perpetuate pulmonary hypertension through inflammation, airway obstruction, and microvasculature damage, all contributing to elevated PVR. Hence, iNO should be used cautiously in this group of patients, only in infants with confirmed pulmonary hypertension on echocardiography. Serial echoes should be performed to guide the duration of iNO, and it should be weaned off as soon as possible when pulmonary hypertension is resolved.

The echocardiographic is essential to assess pulmonary artery pressure (PAP), right ventricular (RV) function, and systemic blood flow (SBF) [74]. Key parameters include estimates of pulmonary artery pressure (PAP), RV function, and systemic blood flow (SBF). The PAP assessment should include the analysis of tricuspid regurgitation jet velocity and PDA shunt velocity, when present, to estimate right ventricular systolic pressure (RVSP) and PAP. Septal wall morphology in the parasternal short-axis view and systolic eccentricity index (EI) provides an objective measure of septal flattening. RV function must be assessed using objective and validated methods, including tricuspid annular plane systolic excursion (TAPSE), fractional area change (FAC), Doppler tissue imaging, and deformation imaging [75]. These echocardiographic assessments enable timely diagnosis and tailored PH-associated pulmonary hypertension in neonates.

#### 6.5.3. Systemic Hypotension

Pulmonary hemorrhage in neonates is often associated with hypotension, driven by blood loss, a reduced preload, hypoxia-induced myocardial dysfunction, and impaired contractility. Hypovolemia and systemic hypoxia can compromise myocardial oxygenation, further reducing cardiac output. Additionally, elevated mean airway pressures (MAPs), often used in PH management, further compromise venous return and preload, exacerbating hypotension. Increased intrathoracic pressure can elevate central venous pressure and predispose individuals to germinal matrix hemorrhage in preterm infants with fragile cerebral vasculature.

Increased intrathoracic pressure also raises the RV afterload, which may impair RV function and exacerbate adverse RV-LV interactions. The echocardiographic assessment is essential to guide clinical management. Key assessments include the following:The assessment of biventricular systolic function (the impact of hemorrhage on contractility).The evaluation of preload and afterload (volume status and vascular resistance).The characterization of intra- and extracardiac shunts (PDA and atrial shunt), influencing pulmonary and systemic flow.Chamber morphology (LV and RV size and hypertrophy, dilation) secondary to volume or pressure overload.The exclusion or inclusion of a diagnosis of congenital heart disease, which may present with hemodynamic instability and contribute to PH [75].

## 7. Management

The effective management of neonatal PH requires a rapid, multidisciplinary approach focused on stabilizing respiratory and hemodynamic status, preventing further bleeding, and treating underlying conditions (Table 2). Key objectives include optimizing oxygenation, minimizing additional pulmonary injury, correcting hemodynamic instability, and managing coagulopathies. Interventions must be prompt and tailored to the infant’s condition, often requiring a combination of strategies related to hemorrhage severity and comorbidities.

### 7.1. Resuscitation and Stabilization

Prompt resuscitation and stabilization are critical. Following standard neonatal resuscitation protocols, hemorrhagic secretions should be carefully cleared from the airway using gentle suctioning, ensuring that the airway remains patent while minimizing additional trauma. If necessary, endotracheal intubation should be performed to secure the airway and facilitate effective ventilation. Gentle suctioning is essential for removing blood clots that may obstruct airflow or impair the visualization of the airway’s anatomy; however, excessive suctioning and frequent disconnection from the ventilator should be avoided to prevent further mucosal injury or the exacerbation of bleeding [24].

### 7.2. Mechanical Ventilation

Mechanical ventilation is central to the management of PH, aiming to support oxygenation while minimizing further lung injury. Increasing the mean airway pressure (MAP)—either by adjusting PEEP on conventional ventilation or by titrating MAP on high-frequency oscillatory ventilation (HFOV)—can help improve oxygenation and tamponade alveolar bleeding. Our approach prioritizes the prevention of hypoxemia and acidosis while avoiding excessive pressures that may exacerbate capillary leaks or worsen pulmonary hemorrhage. Careful balance is required to minimize barotrauma, volutrauma, and further hemodynamic compromises.

High-Frequency Oscillatory Ventilation (HFOV): HFOV is often favored over conventional mechanical ventilation (CMV), although no RCT has confirmed its superiority. Several studies have demonstrated that HFOV can significantly reduce FiO_2_ requirements and improve the oxygenation index in critically ill neonates with massive PH and respiratory failure [8,76,77]. AlKharfy et al. further reported that HFOV effectively promotes adequate ventilation [78]. Additionally, a study by Duval et al. highlighted the potential life-saving benefits of HFOV in cases of severe PH, showing a rapid improvement in oxygenation [79]. Its sustained high distending pressures, which may tamponade alveolar bleeding, reduce pulmonary blood flow, and limit further capillary rupture, make it a valuable lung-protective strategy.Positive End-Expiratory Pressure (PEEP): Trompeter et al. demonstrated that increasing MAP through PEEP optimization alongside acidosis correction, morphine administration, and diuretic (furosemide) can stabilize hemorrhage [5]. Similarly, Bhandari et al. reported benefits from combining increased MAPs with endotracheal epinephrine (1:10,000 at 0.1 mL/kg) and/or 4% cocaine (4 mL/kg) [80]. These strategies aim to improve lung recruitment, stabilize alveolar capillary membranes, and mitigate further hemorrhagic episodes by reducing excessive pulmonary capillary pressure and improving gas exchange.

### 7.3. Surfactant Therapy

While surfactant therapy has been implicated as a potential trigger for PH in some infants, it can be beneficial in most neonates who experience PH once the acute bleeding phase has stabilized [33,81]. Additional doses may be administered post-hemorrhage to enhance lung recruitment and improve gas exchange since alveolar blood inactivates surfactants. Pandit et al. reported an improvement in the oxygenation index following surfactant administration [82], while Amizuka et al. demonstrated a positive effect, showing that 82% of cases achieved a ventilatory index of <0.047 within one hour of surfactant administration. Moreover, no neonates in this study developed BPD or experienced mortality, further suggesting the role of surfactants in the recovery after PH [83].

### 7.4. Blood Product Transfusions and Coagulation Support

Coagulopathies are commonly associated with PH, making hemodynamic stabilization and the correction of clotting abnormalities essential in its management. Blood products are frequently used as adjunctive therapies to replace lost blood volume and restore coagulation function [1,82]. In neonates with PH and severe thrombocytopenia, platelet transfusion decisions should be individualized based on the overall clinical context. Key factors include the severity and location of bleeding, the trajectory of platelet decline, the presence of coagulopathy, and the hemodynamic stability of infants. While high-quality trial data are lacking, current recommendations from expert consensus and guidelines suggest transfusing platelets when counts fall below 50,000/µL in the presence of active bleeding or when invasive procedures are planned [84]. The British Society of Haematology guidelines recommend maintaining hemoglobin levels above 120 g/L in preterm infants requiring ventilatory support or experiencing active bleeding. Fresh-frozen plasma (FFP) may be beneficial for neonates with clinically significant bleeding or abnormal coagulation profiles characterized by prolonged PT or aPTT relative to gestational and postnatal age norms. If PT is elevated, an additional dose of vitamin K may be administered, and fibrinogen levels should be maintained above 1.0 g/dL through fibrinogen supplementation [85]. In addition, several pharmacologic agents can be used to promote hemostasis and stabilize bleeding.

Recombinant Factor VII (rFVIIa): A few case reports have documented the use of intravenous recombinant activated factor VII (rFVIIa) in neonates with life-threatening PH, demonstrating positive outcomes. rFVIIa acts by directly activating the extrinsic coagulation pathway, leading to thrombin generation and fibrin clot formation, which may help control severe bleeding. However, further prospective studies are needed to establish the optimal dosage, timing of administration, and overall efficacy, safety, and tolerability of rFVIIa in both preterm and term neonates [86].Hemocoagulase: Some studies have reported the use of hemocoagulase, a snake venom-derived enzyme, as a hemostatic agent in neonatal hemorrhage. Hemocoagulase promotes blood coagulation by activating prothrombin and accelerating fibrin clot formation. While its use in neonates remains limited, preliminary reports suggest its potential efficacy in controlling pulmonary hemorrhage. Further research is needed to evaluate its safety, optimize dosing, and enhance its overall effectiveness in neonatal care [87,88,89].Antifibrinolytic Agents: Tranexamic acid (TXA) is an antifibrinolytic agent that prevents fibrin clot degradation by inhibiting plasminogen activation. It has been used intravenously in some cases of neonatal hemorrhage, including PH, to stabilize existing clots and reduce ongoing bleeding [90]. While TXA has shown promise in adult and pediatric populations [91], its use in neonates remains limited, and further research is needed to establish its safety, optimize dosing, and improve overall efficacy in preterm and critically ill infants.Ankaferd Blood Stopper (ABS): Two cases from a report described the successful use of ABS in treating massive PH in neonates. The first case involved a term male newborn at 38 3/7 weeks of gestation, while the second case involved a late preterm infant at 33 6/7 weeks. Both neonates experienced severe PH, and ABS was administered directly via the endotracheal tube. In both cases, the hemorrhage ceased immediately following ABS administration, highlighting its potential as an emergency hemostatic intervention in neonatal PH. However, further studies are necessary to evaluate its safety, optimize dosing, and apply it to broader clinical applications, especially in premature infants where there is no reported evidence or experience [92].

### 7.5. Endotracheal Epinephrine

Although not standard therapy, endotracheal epinephrine may be considered in life-threatening cases of PH, particularly during resuscitation or when bleeding is refractory. Epinephrine administered via the endotracheal tube (ETT) induces localized pulmonary vasoconstriction to help control the hemorrhage. While effective, this approach carries potential risks, including airway or vocal cord ischemia, local tissue necrosis, arrhythmias, systemic vasoconstriction, hyperglycemia, lactic acidosis, and hypertension. A study by Chen et al. reported significantly higher survival rates (80% vs. 18.2%) with direct intratracheal catheter administration (1:10,000 epinephrine, dose 0.3 to 1.0 mL/kg (0.03 to 0.1 mg/kg) compared to ETT connector delivery [17]. However, further studies are needed to define optimal dosing and safety in this context.

### 7.6. Tolazoline

In one case report, tolazoline was administered to a full-term newborn with neonatal encephalopathy and pulmonary hemorrhage. The infant received a 6 mg bolus followed by an intravenous infusion of tolazoline, leading to a rapid improvement in oxygenation within three minutes. This improvement facilitated the discontinuation of ventilatory support 16 h later. The newborn ultimately survived without any apparent long-term disability, suggesting the potential role of tolazoline in the management of severe neonatal pulmonary hypertension. However, further studies are needed to assess its safety and efficacy in broader neonatal populations [93].

In summary, during the “golden hour”, the primary goal is to minimize factors that may precipitate or exacerbate PH. This involves providing balanced respiratory support to tamponade alveolar bleeding while avoiding overinflation and impaired cerebral venous return, which could increase the risk of intraventricular hemorrhage (IVH). Normothermia should be maintained, fluid balance carefully optimized, and blood product replacement judiciously used based on hematologic parameters. Surfactants should be administered in the case of hypoxic respiratory failure, given its inactivation by intra-alveolar blood. If bleeding persists or hemodynamic instability is evident, the early evaluation of the cardiovascular phenotype through thorough clinical assessment and targeted neonatal echocardiography is recommended, followed by tailored hemodynamic interventions. In rare, life-threatening, and refractory cases, endotracheal epinephrine applied at the tip of the ETT using another catheter inserted with the ETT may be considered as a last resort to induce localized vasoconstriction and achieve temporary hemostasis.

## 8. Prognosis

The prognosis of PH in preterm infants depends on its severity and associated complications [14]. Chen et al. reported a significantly higher incidence of severe IVH (grades 3 or 4) in infants with massive PH, with all affected neonates developing IVH either concurrently or shortly after. The pathogenesis of IVH is complex and multifactorial and is described previously in this study [17]. Similarly, Pandit et al. found that neonates with PH were three times more likely to develop major IVH [11]. Concomitant infants experiencing severe PH had a significantly higher risk of death or neurodevelopmental impairment among survivors [62].

This review is narrative in nature and does not follow a formal systematic search or critical appraisal process. Articles were selected based on their relevance to PH in preterm infants and the authors’ clinical and academic expertise. A standardized method for assessing study quality was not applied. As such, the findings presented should be interpreted as a synthesis of current knowledge and expert opinion rather than a definitive or exhaustive analysis.

## 9. Future Directions

Despite progress, many gaps remain in our understanding and management of PH in preterm infants. The complex interplay between hemodynamic, coagulation, and genetic predisposition is not fully elucidated. Risk factors such as RDS, PDA, and surfactant use need better characterization, especially in combination. Diagnostic tools like LUS and echocardiography show promise but lack standardization implementation. Additional predictive strategies for PH must be explored, such as heart rate variability analysis or artificial intelligence models integrating multiple physiological signals and near-infrared spectroscopy by establishing normative values specific to preterm populations and assessing their impact on clinical decision-making and outcomes. Therapeutic strategies also warrant further investigation. While HFOV and surfactant therapy are commonly utilized, optimal timing, dosing, and long-term effects remain unclear. Adjunctive therapies—such as antifibrinolytics, rFVII, hemocoagulase, and ABS—require further validation. Preventive approaches, including antenatal corticosteroids and prophylactic indomethacin, also warrant targeted trials. Finally, the long-term impact of PH on survivors remains poorly characterized. Future studies should address the neurodevelopmental and cardiopulmonary sequelae and the broader economic and psychosocial burden on families. Understanding the trajectory of PH survivors is crucial to refine follow-up strategies and optimize long-term care. Addressing these gaps will be essential to improving both survival and short- and long-term outcomes in this vulnerable population.

## 10. Conclusions

PH in preterm infants remains a major clinical challenge due to its complex and multifactorial pathogenesis. Despite advancements in neonatal care, PH continues to be associated with high mortality and severe long-term neurodevelopmental impairment. Emerging evidence highlights the importance of early recognition through clinical signs, laboratory markers, and imaging, particularly lung ultrasound. Management strategies such as high-frequency oscillatory ventilation, higher PEEP levels, surfactant administration, and hemostatic support play key roles during acute care. Future research should focus on refining preventive strategies, validating early diagnosis tools, and exploring novel therapies to improve survival and long-term outcomes in this high-risk population.

## Figures and Tables

**Figure 1 biomedicines-13-01744-f001:**
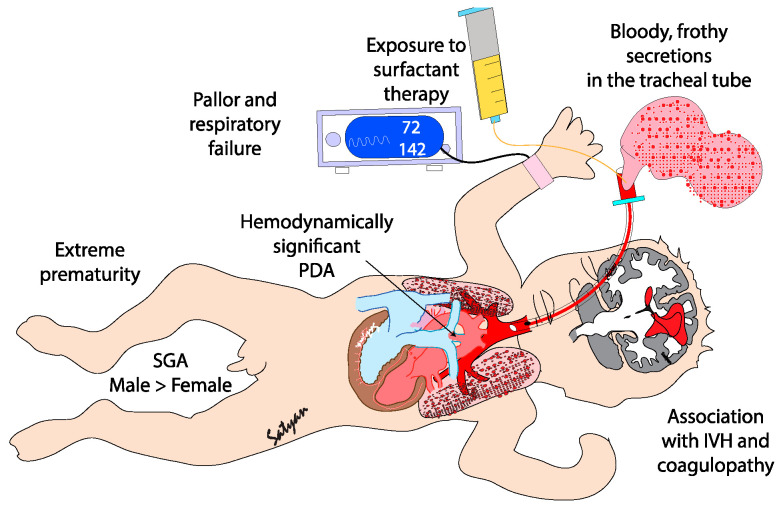
Clinical features of pulmonary hemorrhage in premature infants. Extremely premature infants, especially those small for gestation (SGA) with a hemodynamically significant patent ductus arteriosus that have been exposed to a surfactant, are at risk for pulmonary hemorrhage. Coagulopathy and severe intraventricular hemorrhage (IVH) may be associated with pulmonary hemorrhage. Copyright Satyan Lakshminrusimha.

**Figure 2 biomedicines-13-01744-f002:**
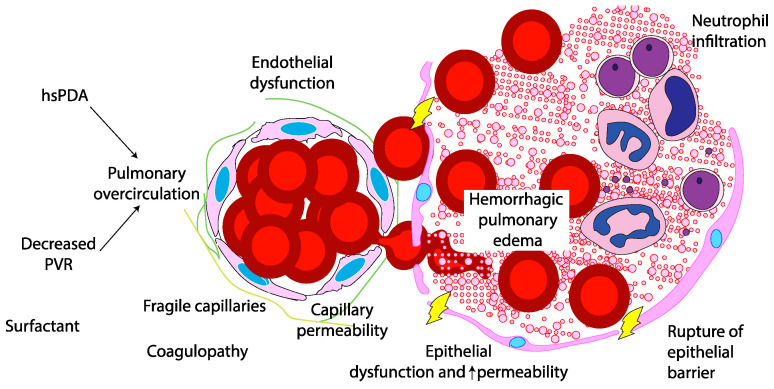
The pathogenesis of pulmonary hemorrhage. An airspace–capillary interface can be disrupted by the rupture of the epithelial barrier and endothelial dysfunction with increased capillary permeability. Surfactant therapy with decreased pulmonary vascular resistance (PVR) coupled with a hemodynamically significant patent ductus arteriosus (PDA) can increase pulmonary blood flow, disrupt fragile capillaries, and result in neutrophil and red blood cell infiltration and edema in the airspace. Copyright Satyan Lakshminrusimha.

**Figure 3 biomedicines-13-01744-f003:**
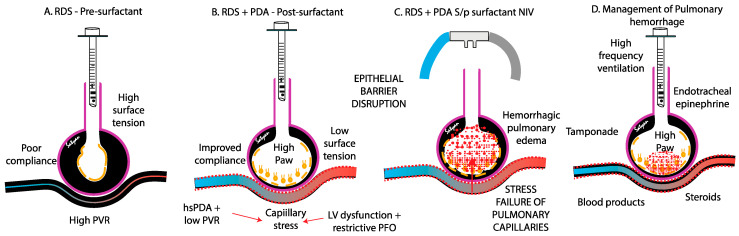
The potential pathophysiology and dynamic interactions contributing to pulmonary hemorrhage in preterm infants with respiratory distress syndrome (RDS) and hemodynamically significant patent ductus arteriosus (hsPDA). (**A**) Before surfactant administration, preterm infants with RDS have high surface tension in the alveolus with high mean airway pressures due to poor compliance and low pulmonary blood flow due to high pulmonary vascular resistance (PVR). (**B**) Following surfactant administration, compliance increases, surface tension decreases, and pulmonary blood flow increases (especially with a hsPDA). Concomitant left ventricular (LV) dysfunction with a restrictive patent foramen ovale (PFO) induces capillary stress and the disruption of the epithelial barrier, causing hemorrhagic pulmonary edema. (**C**) Extubation with non-invasive ventilation (NIV) and the loss of alveolar pressure may potentially contribute to an increase in edema. Airspace tamponades with high mean airway pressures (with PEEP or high-frequency ventilation), endotracheal epinephrine, blood products to correct coagulopathy and anemia, and steroids may be beneficial in management (**D**). Copyright Satyan Lakshminrusimha.

**Figure 4 biomedicines-13-01744-f004:**
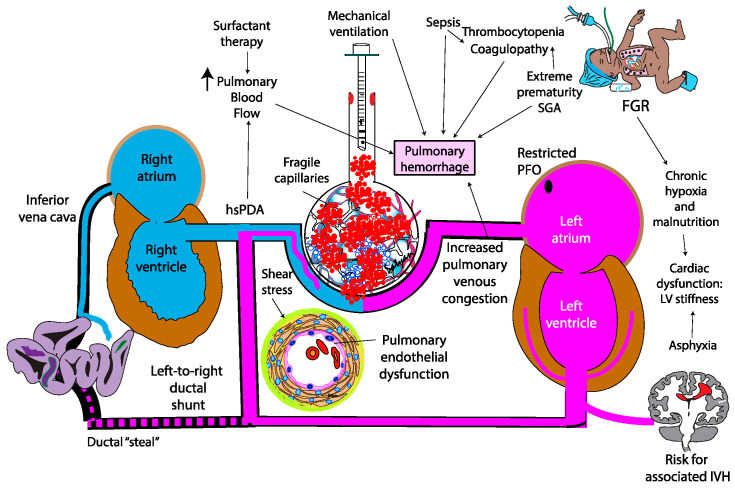
The contributing factors and pathophysiology of pulmonary hemorrhage in preterm infants. SGA—small for gestational age; FGR—fetal growth restriction; hsPDA—hemodynamically significant patent ductus arteriosus; LV—left ventricle; IVH—intraventricular hemorrhage. Increased pulmonary blood flow due to hsPDA and low PVR combined with restricted outflow due to small PFO and LV dysfunction increases capillary stress. Copyright Satyan Lakshminrusimha.

**Figure 5 biomedicines-13-01744-f005:**
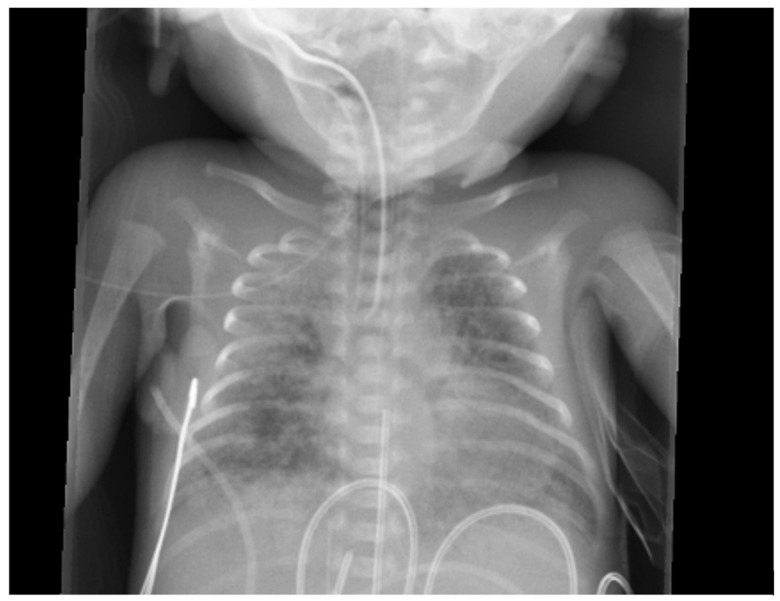
A chest X-ray of a premature baby with pulmonary hemorrhage showing bilateral fluffy opacities reflecting localized alveolar hemorrhage.

**Figure 6 biomedicines-13-01744-f006:**
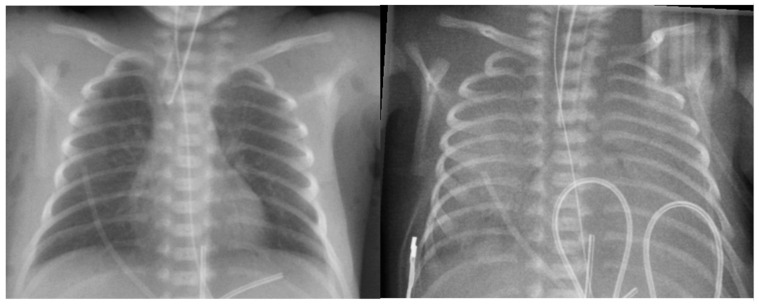
A chest X-ray of a baby with massive pulmonary hemorrhage. Left: Before pulmonary hemorrhage. Right: After massive pulmonary hemorrhage: a “white out” lung appearance can be seen due to extensive pulmonary flooding.

**Table 1 biomedicines-13-01744-t001:** Risk and protective factors of pulmonary hemorrhage.

Risk Factors	Protective Factors
Respiratory distress syndrome—A sudden drop in PVR after surfactant therapy and pressure from the mechanical ventilator leads to capillary trauma. Patent ductus arteriosus and pulmonary over circulation—Pulmonary edema and capillary rupture. Stress failure of pulmonary capillaries—The disruption of the endothelial barrier, allowing hemorrhagic fluid to leak into alveoli. Left ventricular stiffness and diastolic properties—Myocardial workload and afterload mismatch. Genetic factors—HHT: impaired vascular integrity; hemophilia: bleeding disorder Infection and sepsis—Profound microvascular injury and inflammation. Fetal growth restriction—Coagulopathy, thrombocytopenia, platelet dysfunction, and cardiovascular dysfunction. Possible factors—Congenital heart disease, asphyxia, hypothermia, and lower coagulation factors in neonatal populations.	Antenatal glucocorticoids—Enhancing surfactant production and stabilizing the pulmonary vasculature. Prophylactic indomethacin—Lower incidence of early, severe PH (within the first week of life).

Abbreviations: PVR (pulmonary vascular resistance); HHT (hereditary hemorrhagic telangiectasia); PH (pulmonary hemorrhage).

**Table 2 biomedicines-13-01744-t002:** Management of pulmonary hemorrhage in neonates.

Initial Assessment and Stabilization	- Assess for clinical signs of PH (bloody secretions, hypoxemia, hypotension). Exclude traumatic causes of bleeding (position of ETT). - Consider early intubation and initiation of mechanical ventilation if not already intubated. - Perform chest radiography and consider lung ultrasound to assess alveolar infiltrates. - Monitor continuously with pulse oximetry, blood pressure measurement, and blood gas analysis.
Mechanical Ventilation	- Secure the airway. - Increase the PEEP level to prevent further hemorrhage and improve oxygenation. - Initiate or transition to HFOV to minimize alveolar collapse and optimize gas exchange. - Employ a gentle ventilation strategy to reduce barotrauma and protect fragile lung tissue.
Surfactant Therapy	- Administer additional surfactant doses once the bleeding has stabilized. - Monitor for hemodynamic changes following surfactant administration.
Blood Product Transfusions	- Administer packed red blood cells to maintain hemoglobin levels. - Administer fresh frozen plasma (FFP) for coagulation factor replacement in cases of coagulopathy. - Give cryoprecipitate to replenish fibrinogen and other clotting factors if levels are low. - Perform platelet transfusions if thrombocytopenia is present
Coagulation Support	- Administer vitamin K if ongoing DIC is detected.
Endotracheal Epinephrine	- Consider administering endotracheal epinephrine if encountering persistent and uncontrolled bleeding or during resuscitation.
Inotropic Drug or Vasopressor	- Consider using an inotropic drug or vasopressor to help stabilize the hemodynamic status.
Steroids	- Consider steroids in cases of potential adrenal insufficiency.

Abbreviations: ETT (endotracheal tube); PEEP (positive end-expiratory pressure); HFOV (high-frequency oscillator ventilation); DIC (disseminated intravascular coagulopathy).

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
