# Peer review of "Pulmonary Hemorrhage in Premature Infants: Pathophysiology, Risk Factors and Clinical Management"

_biomedicines, 2025, doi:10.3390/biomedicines13071744_

Round 1
Reviewer 1 Report
Comments and Suggestions for Authors
- Mention about the type of review
- Elaborate the introduction part and mention about the evolution of knowledge regarding Pulmonary haemorrhage in preterm infants. What is the knowledge gap you want explore?
- Make section of methods with PRISMA flow diagram
- It’s looking like a book chapter, no need to write about prognosis
- There must be limitation segment in discussion part
- How do you envision LUS being integrated into routine neonatal intensive care workflows, particularly in resource-limited settings?
- What specific diagnostic criteria or algorithms would you recommend to differentiate PH from severe sepsis cases?
- What first-line interventions or protocols you advocate for during the 'golden hour'.
- In neonates with PH and severe thrombocytopenia, how do you balance the risks of platelet transfusion
- As mentioned role of lung ultrasound for early diagnosis, but its interpretation requires expertise. How might emerging technologies like AI-assisted LUS analysis or point-of-care biomarkers help standardize and expedite PH detection in routine NICU practice
- Are there specific ventilator settings or hemodynamic parameters you prioritize when stabilizing infants with acute PH?
Author Response
Comment 1 Mention about the type of review
Response 1: We have clarified by adding the mention “narrative” at the end of the introduction on line 69: “This narrative review synthesizes current evidence, addresses key diagnostic and therapeutic challenges, and proposes future directions to improve outcomes in this vulnerable population”
Comment 2 Elaborate the introduction part and mention about the evolution of knowledge regarding Pulmonary haemorrhage in preterm infants. What is the knowledge gap you want explore?
Response 2: We would like to thank the reviewer for their comment. We have added the following sentence to provide broader context and emphasize the evolving understanding of PH in preterm infants (line 60, page 2): “Over the past decades, our understanding of PH in preterm infants has evolved from a narrow focus on vascular fragility to a broader recognition of its multifactorial pathophysiology. Contributing factors include pulmonary hemodynamic disturbances, surfactant-induced changes, cardiac immaturity, prenatal influences, impaired left-sided cardiac compliance, and genetic or inflammatory predispositions. Despite these insights, significant gaps remain—particularly in standardizing diagnostic criteria, clarifying the interplay among known risk factors, and integrating early detection tools such as lung ultrasound, heart rate variability analysis, and targeted neonatal echocardiography. Additionally, effective prevention and management strategies are yet to be clearly defined. This narrative review synthesizes current evidence, addresses key diagnostic and therapeutic challenges, and proposes future directions to improve outcomes in this vulnerable population.”
Comment 3 Make section of methods with PRISMA flow diagram
Response 3: Thank you for your valuable feedback. We appreciate your suggestion to include a PRISMA flow diagram. However, this manuscript is a narrative review rather than a systematic review or meta-analysis; therefore, a PRISMA flow diagram is not applicable and was not included.
Comment 4 It’s looking like a book chapter, no need to write about prognosis
Response 4: Thank you for your thoughtful comment. We included the prognosis section to offer a more comprehensive overview and to help readers appreciate the clinical significance of the condition. We believe this addition is particularly valuable for clinicians and trainees. That said, we have carefully reviewed the section to ensure it remains concise, focused, and relevant to the overall scope of the review.
Comment 5 There must be limitation segment in discussion part
Response 5: Thank you for your comment. As this is a narrative review, it does not include a traditional discussion section. However, we agree that acknowledging the limitations is important. We propose adding the following paragraph on line 670 page 17 to address this: “This review is narrative in nature and does not follow a formal systematic search or critical appraisal process. Articles were selected based on their relevance to PH in preterm infants and the authors’ clinical and academic expertise. A standardized method for assessing study quality was not applied. As such, the findings presented should be interpreted as a synthesis of current knowledge and expert opinion, rather than a definitive or exhaustive analysis.”
Comment 6 How do you envision LUS being integrated into routine neonatal intensive care workflows, particularly in resource-limited settings?
Response 6: Thank you for this important question. We view lung ultrasound (LUS) as a promising tool for future research into the diagnosis and monitoring of neonatal pulmonary hemorrhage, particularly in resource-limited settings where access to advanced imaging may be restricted. Its portability, low cost, and absence of radiation make it well-suited for evaluating its performance in diagnostic assessment and the monitoring of preventive or therapeutic strategies. However, its routine clinical use in pulmonary hemorrhage remains limited by a lack of high-quality evidence.
Preliminary studies suggest that certain LUS features—such as lung consolidation, air or fluid bronchograms, pleural effusion, B-lines, and shred signs—are frequently observed in neonates with established PH. Liu et al. (2023) and Ren et al. (2017) described in small retrospective studies (n = 42 and n = 57, respectively) relatively consistent ultrasound findings in confirmed cases, with high reported sensitivities for signs like the shred sign. However, these findings are not specific to pulmonary hemorrhage and may also occur in pneumonia or meconium aspiration. Importantly, in both studies, imaging was performed after clinical diagnosis of hemorrhage, limiting conclusions about predictive value or applicability for early detection and therapeutic guidance. These studies were also limited by small sample sizes, subjective diagnostic criteria, and lack of generalizability to other clinical contexts—especially low-income settings. No trial to date has assessed whether LUS can meaningfully predict pulmonary hemorrhage, support accurate diagnosis, or guide management in ways that improve outcomes such as survival, severe IVH, or pulmonary morbidity. Moreover, given the overlap of LUS findings with more benign conditions, premature adoption without robust evidence could lead to overdiagnosis, unnecessary treatment, or diagnostic anchoring, potentially causing harm. While LUS remains an appealing tool with significant potential, especially in low-resource settings, its integration into clinical pathways must be supported by prospective research and trials assessing its value in prevention, diagnosis, and therapeutic titration.
We have added a statement to the manuscript reflecting these considerations and the need for further research on line 425 page 12: “LUS is a promising bedside modality with potential utility in the diagnosis and monitoring of PH in neonates. Its radiation-free, low-cost nature makes it particularly appealing in resource-limited settings. Preliminary studies suggest that certain LUS features—such as lung consolidation, air or fluid bronchograms, pleural effusion, B-lines, and shred signs—are frequently observed in neonates with established PH. Liu et al. (2023) and Ren et al. (2017) described in retrospective studies (n = 42 and n = 57, respectively) relatively consistent ultrasound findings in confirmed cases, with high reported sensitivities for signs like the shred sign (refers to irregular, jagged or serrated border appearance of the pleural-lung interface, described as the appearance of "torn paper".). However, these findings are not specific to PH and may also occur in pneumonia or meconium aspiration. Importantly, in both studies, imaging was performed after clinical diagnosis of hemorrhage, limiting conclusions about predictive value or applicability for early detection and therapeutic guidance. These studies were also limited by small sample sizes, subjective diagnostic criteria, and lack of generalizability to other clinical contexts—especially low-income settings. No trial to date has assessed whether LUS can meaningfully predict pulmonary hemorrhage, support accurate diagnosis, or guide management in ways that improve outcomes such as survival, severe IVH, or pulmonary morbidity. Moreover, given the overlap of LUS findings with more benign conditions, premature adoption without robust evidence could lead to overdiagnosis, unnecessary treatment, or diagnostic anchoring, potentially causing harm. While LUS remains an appealing tool with significant potential, especially in low-resource settings, its integration into clinical pathways must be supported by prospective research and trials assessing its value in prevention, diagnosis, and therapeutic titration.”
We removed “ LUS is increasingly recognized as a potential tool for pulmonary conditions in neonates, including PH. As a non-invasive, radiation-free, and bedside-performable technique, LUS is particularly advantageous for critically ill neonates requiring frequent assessment. Studies have demonstrated high sensitivity and specificity for detecting various pulmonary pathologies. A study by Ren et al. highlighted the reliability of LUS for recognizing PH[62].”
Comment 7 What specific diagnostic criteria or algorithms would you recommend to differentiate PH from severe sepsis cases?
Response 7: Thank you for this thoughtful question. At present, there is no studied algorithm to distinguish pulmonary hemorrhage from severe neonatal sepsis. Differentiation relies on clinical judgment, with attention to the timing and character of deterioration. PH typically presents with abrupt respiratory decompensation and frank pulmonary bleeding, while sepsis more often follows a course with systemic signs such as temperature instability, hypotension, coagulopathy, and metabolic acidosis. However, these conditions are not mutually exclusive—pulmonary hemorrhage may occur as part of a broader systemic inflammatory response, particularly in the setting of overwhelming infection with thrombocytopenia, hepatic failure, hemodynamic instability and disseminated intravascular coagulation, making diagnosis even more challenging. Further research is needed to develop and validate clinical frameworks that can distinguish primary PH from pulmonary involvement secondary to sepsis. Future studies should explore the integration of bedside imaging (e.g., lung ultrasound, targeted neonatal echocardiography), laboratory markers of inflammation and coagulation, and hemodynamic parameters into diagnostic algorithms. A better understanding of the pathophysiological interplay between SIRS and PH will be essential to refine diagnostic precision and guide targeted interventions. We have added a statement to the manuscript reflecting these diagnostic challenges and the need for research into more robust differentiation strategies: “Differentiating PH from severe sepsis remains clinically challenging, particularly when both conditions coexist as part of a systemic inflammatory response. Further research is needed to develop integrated diagnostic frameworks combining imaging, laboratory, and hemodynamic data to improve accuracy and guide management when there is diagnostic uncertainty between these two conditions.”- line 292 page 7
Comment 8 What first-line interventions or protocols you advocate for during the 'golden hour'.
Response 8: We would like to thank the reviewer for giving us the opportunity to provide the readership with a guided approach. We have added this mention to the text at the end of the management section line 648 page 17 : “In summary, during the ‘golden hour,’ the primary goal is to minimize factors that may precipitate or exacerbate PH. This involves providing balanced respiratory support to tamponade alveolar bleeding while avoiding overinflation and impaired cerebral venous return, which could increase the risk of intraventricular hemorrhage. Normothermia should be maintained, fluid balance carefully optimized, and blood product replacement used judiciously based on hematologic parameters. Surfactant should be administered in the setting of hypoxic respiratory failure, given its inactivation by intra-alveolar blood. If bleeding persists or hemodynamic instability is evident, early evaluation of the cardiovascular phenotype through thorough clinical assessment and targeted neonatal echocardiography is recommended, followed by tailored hemodynamic interventions. In rare, life-threatening and refractory cases, endotracheal epinephrine applied at the tip of the ETT using another catheter inserted with the ETT may be considered as a last resort to induce localized vasoconstriction and achieve temporary hemostasis”.
Comment 9 In neonates with PH and severe thrombocytopenia, how do you balance the risks of platelet transfusion
Response 9: Current recommendations, including published guidelines on platelet transfusion support in neonates, support the use of a threshold of <50,000/µL for transfusion in the setting of active bleeding or prior to invasive procedures (see references below). However, the evidence base remains limited, and further research is needed to better define optimal transfusion thresholds in this high-risk population with active bleeding. We have added the following statement to the manuscript line 583 page 15: “In neonates with PH and severe thrombocytopenia, platelet transfusion decisions should be individualized based on the overall clinical context. Key factors include the severity and location of bleeding, the trajectory of platelet decline, the presence of coagulopathy, and the infant’s hemodynamic stability. While high-quality trial data are lacking, current recommendations from expert consensus and guidelines suggest transfusing platelets when counts fall below 50,000/µL in the presence of active bleeding or when invasive procedures are planned ”.
Comment 10 As mentioned role of lung ultrasound for early diagnosis, but its interpretation requires expertise. How might emerging technologies like AI-assisted LUS analysis or point-of-care biomarkers help standardize and expedite PH detection in routine NICU practice
Response 10: To address this particular comment, we have added this mention to the LUS section-line 456 page 12: “The concept of AI-assisted LUS interpretation holds promise for improving early detection and standardization of PH diagnosis, especially in settings lacking expert imaging interpretation. If properly developed, these technologies could enhance diagnostic consistency, support earlier intervention, and reduce reliance on advanced imaging infrastructure. AI tools are still in early developmental stages, and their application to PH requires collaborative work in bioengineering and neonatology. Prospective studies will be essential to design, validate, and evaluate the accuracy and clinical impact of these technologies within diagnostic frameworks.”
Comment 11 Are there specific ventilator settings or hemodynamic parameters you prioritize when stabilizing infants with acute PH?
Response 11: We thank the reviewer for this important comment, and we have added this mention on line 543 page 14: “Increasing mean airway pressure (MAP)—either by adjusting PEEP on conventional ventilation or by titrating MAP on high-frequency oscillatory ventilation (HFOV)—can help improve oxygenation and tamponade alveolar bleeding. Our approach prioritizes the prevention of hypoxemia and acidosis, while avoiding excessive pressures that may exacerbate capillary leak or worsen pulmonary hemorrhage. Careful balance is required to minimize barotrauma, volutrauma, and further hemodynamic compromise.”
Reviewer 2 Report
Comments and Suggestions for Authors
This is a very-well thought through, vey comprehensive review about pulmonary hemmorhage. Apart from various typos which will hopefully be solved in post-processing, there is little I can fault this manuscript with.
One issue I think needs further clarification is using reference 28 on line 142, due to the fact that the reference is quite old and might not be applicable to those neonates of extremely low GA, which might have not survived back in 1993.
I believe on line 187, the word „screening” should be replaced by „treatment”. The recommendation of early treatment of the PDA in the following paragraph contradicts the recommendations for expectant therapy issued recently by the American Academy of Pediatrics – this is something to be better clarified.
Minor issues I would point out are linked to the figures and tables:
- Figure 1 can be sized down (the font is unnecessarily large) – this may solve the following issue
- Figure 2 needs to be placed differently so that the figure itself and the legend are on the same page
- Both tables can be sized down by using a small font and smaller spacing. Also, although they are a synthesis of the text, they should contain all the abbreviations, so they may be read independently
- Figure 5 is also unnecessarily large and I would like a more comprehensive explanation for the image in the legend
Author Response
Comments1: One issue I think needs further clarification is using reference 28 on line 142, due to the fact that the reference is quite old and might not be applicable to those neonates of extremely low GA, which might have not survived back in 1993.
Response 1: We would like to thank the reviewer for this comment. The original sentence on line 142 stated: “A meta-analysis of surfactant therapy has shown a modest increase in the risk of PH.” We have now revised the text to better reflect the limitations of the existing evidence – line 150 page 4: “A meta-analysis of surfactant therapy published in 1993 reported a modest increase in the risk of pulmonary hemorrhage. However, this study predates the routine survival of extremely preterm infants born at <25 weeks’ gestation and may not be generalizable to the current population. Updated studies in this specific subgroup are needed to clarify the risk.”
Comment 2: I believe on line 187, the word „screening” should be replaced by „treatment”. The recommendation of early treatment of the PDA in the following paragraph contradicts the recommendations for expectant therapy issued recently by the American Academy of Pediatrics – this is something to be better clarified.
Response 2 Thank you for pointing out. The sentence 187 was reading: “Centers that use high frequency ventilation and early hemodynamic screening for PDA report lower incidence of PH (figure 3).” We have corrected for “Centers that use high frequency ventilation and early hemodynamic treatment for PDA report lower incidence of PH (figure 3). However, as outlined in a recent meta-analysis and a review by our group the early use of medications to accelerate ductal closure in extremely preterm infants does not appear to confer short- or long-term benefit. This practice has been increasingly discouraged, including in the most recent guidelines from the American Academy of Pediatrics”- line 199 page 5
Comment 3: Minor issues I would point out are linked to the figures and tables:
- Figure 1 can be sized down (the font is unnecessarily large) – this may solve the following issue
- Figure 2 needs to be placed differently so that the figure itself and the legend are on the same page
- Both tables can be sized down by using a small font and smaller spacing. Also, although they are a synthesis of the text, they should contain all the abbreviations, so they may be read independently
- Figure 5 is also unnecessarily large and I would like a more comprehensive explanation for the image in the legend
Response: We would like to thank the reviewer for these recommendations. We have modified as recommended:
- Figure 1 is smaller now – page 2.
- Figure 2 is now on the same page as its legend- page 3
- Both tables are smaller with abbreviations explained- page 10, 18
- Figure 5 is now smaller with more explanation in the legend- page 11
Round 2
Reviewer 1 Report
Comments and Suggestions for Authors
Nicely revised the manuscript. Congratulations.
Reviewer 2 Report
Comments and Suggestions for Authors
Thank you to the authors for making the suggested changes, I believe this has improved their work and it can now be published as is, in my oppinion.